# Implementation of Nurse Navigation Improves Rate of Molecular Tumor Testing for Ovarian Cancer in a Gynecologic Oncology Practice

**DOI:** 10.3390/cancers15123192

**Published:** 2023-06-15

**Authors:** Taylor A. Rives, Heather Pavlik, Ning Li, Lien Qasrawi, Donglin Yan, Justine Pickarski, Charles S. Dietrich, Rachel W. Miller, Frederick R. Ueland, Jill M. Kolesar

**Affiliations:** 1Division of Gynecologic Oncology, Department of Obstetrics and Gynecology, University of Kentucky, Lexington, KY 40536, USA; taylor.rives@uky.edu (T.A.R.); heather.schroer@uky.edu (H.P.); charles.dietrich@uky.edu (C.S.D.); raware00@uky.edu (R.W.M.); frederick.ueland@uky.edu (F.R.U.); 2Markey Cancer Center, University of Kentucky, Lexington, KY 40536, USA; justine.cooper@uky.edu; 3Department of Biostatistics, College of Public Health, University of Kentucky, Lexington, KY 40536, USA; ning.li@uky.edu (N.L.); donglin.yan@uky.edu (D.Y.); 4College of Pharmacy, University of Kentucky, Lexington, KY 40536, USA; lqa222@uky.edu

**Keywords:** ovarian cancer, nurse navigation, somatic tumor testing, next generation sequencing, molecular tumor testing, germline testing

## Abstract

**Simple Summary:**

The National Comprehensive Cancer Network recommends somatic tumor testing for all patients with epithelial ovarian cancer, as harboring specific mutations has implications for therapy. Targeted panel testing includes some of these genes; however, Next Generation Sequencing is preferred for the complete gene complement. The rate of guideline-concordant molecular tumor testing is low; therefore, we implemented a nurse navigator to improve rates. This study confirms that education sessions, consensus building, and NN implementation improves the rate and timeliness of molecular testing in patients with epithelial ovarian cancer. Next Generation Sequencing revealed a higher rate of actionable mutations that would have been missed using targeted panel testing alone.

**Abstract:**

Purpose: The purpose of this study was to assess the impact of implementing a Nurse Navigator (NN) to improve the rate and timeliness of molecular tumor testing. Methods: This is an evaluation of the impact of education sessions, consensus building, and NN implementation for molecular tumor testing in patients with epithelial ovarian cancer. The NNs’ responsibilities included attending tumor boards and ensuring Next Generation Sequencing (NGS) is ordered, reviewed, and coordinated for appropriate patients. Results: NNs significantly improved NGS testing rates from 35.29% to 77.27%, *p* = 0.002. Ordering a targeted panel test (TPT) was the most common reason for not ordering NGS in the pre-NN cohort (13/22, 59%). The total turnaround time for testing was reduced after the introduction of NNs from 145.2 days to 42.8 days, *p* < 0.0001. The post-NN group had a significantly higher rate of actionable mutations identified for the recurrent setting [67.6% versus 20.8% (*p* = 0.0005)] and a trend towards a higher rate of actionable mutations identified in the frontline setting [41.2% versus 33.3% (*p* = 0.41)]. Conclusion: NNs significantly improved somatic tumor testing rates and timeliness for patients with ovarian cancer. Discontinuing TPT in favor of NGS revealed a higher rate of actionable tumor mutations that would have been missed with TPT alone.

## 1. Introduction

Ovarian cancer is the leading cause of gynecologic cancer-related death in the United States. In 2023, there are estimated to be 19,710 new ovarian cancer diagnoses and 13,270 ovarian cancer-related deaths in the United States [1]. Epithelial ovarian cancer has well-known associations with specific germline and somatic tumor mutations and harboring these mutations can impact treatment decisions. Approximately 18% of ovarian cancer is associated with germline mutations, most attributed to *BRCA1/2* mutations [2]. All patients with epithelial ovarian cancer should undergo germline genetic testing for *BRCA1/2* mutations and other mutations associated with ovarian cancer due to the high rate of hereditary disease in this patient population and the availability of targeted therapies for patients with specific mutations [3,4,5]. 

In 2020, the American Society of Clinical Oncology (ASCO) recommended molecular tumor testing for *BRCA1/2* mutations in all epithelial ovarian cancer patients who were germline negative and recommended testing for mismatch repair (MMR) deficiency in patients with clear cell, endometrioid, or mucinous cancers [3]. However, relying on only germline testing could result in missing 5% of somatic mutations, whereas only relying on somatic tumor testing could result in missing 5% of germline mutations [3]. Currently, the National Comprehensive Cancer Network (NCCN) recommends germline and somatic tumor testing for all patients with epithelial ovarian cancer at the time of diagnosis and recommends that somatic testing includes the evaluation of, at a minimum: *BRCA1/2* mutations, homologous recombination deficiency (HRD) status, microsatellite instability (MSI), tumor mutation burden (TMB), and *NTRK*, *BRAF*, *FRα*, and *RET*, as mutations in these genes have implications for therapies [5]. HRD refers to deficiencies in homologous recombination pathways, which are associated with mutations in *ATM*, *BARD1*, *BRIP1*, *CHEK1*, *CHEK2*, *FANCA*, *FANCC*, *MRE11A*, *NBS1*, *PALB2*, *RAD50*, *RAD51*, *RAD51B*, *RAD51C*, *RAD52*, and *RAD54L*, as well as a high loss of heterozygosity (LOH) score (≥16) [6]. Targeted somatic testing panels include some of these genes; however, Next Generation Sequencing (NGS) is preferred to evaluate the full complement of genes as well as TMB, LOH, and MSI. Over 50% of ovarian cancer patients have a targetable mutation, as there are several FDA-approved targeted therapies along with many others are being developed or evaluated in clinical trials [7]. 

In addition, despite recommendations for germline and somatic tumor testing for patients with epithelial ovarian cancer, the rate of genetic testing is approximately 30% and the rate of molecular testing (germline and somatic) is only 22.9% [8,9,10]. Nurse navigation has demonstrated efficacy in improving both the timeliness of cancer care and patient outcomes [11,12,13,14]. In one breast cancer clinic, the implementation of nurse navigation led to significant improvements in molecular tumor testing rates from 26% to 88% and improvements in the timeliness of testing by decreasing the ordering and test reporting turnaround time by 15 days and 18 days, respectively [12]. In September of 2021, Markey Cancer Center (MCC) implemented a nurse navigator to improve the rates of guideline-concordant molecular tumor testing for ovarian cancer. 

The purpose of this study was to assess the impact of a nurse navigator on molecular testing rates and the timeliness of testing. 

## 2. Materials and Methods

This study is a single-institution evaluation of the impact of a nurse navigator intervention in patients with ovarian cancer treated at Markey Cancer Center in Lexington, Kentucky. The intervention consisted of an initial education session with gynecological surgeons regarding the NCCN testing guidelines, specifically the mutations that should be tested and the mutations assessed by commonly used, commercially available TPT, and NGS tests. Following the educational session, a facilitated discussion was held to reach a consensus on the timing of testing and preferred test. It was decided, per guidelines, that all patients would be tested for germline and somatic mutations at diagnosis with a comprehensive NGS test. The nurse navigator was also introduced at this session with an overview of proposed responsibilities. Subsequent meetings were held to finalize the workflow and responsibilities of the nurse navigator and genetic counselor. The study team agreed that the nurse navigator was responsible for the following: (1) attending gynecologic oncology tumors boards; (2) recommending and ordering NGS when appropriate; (3) interacting with clinical staff, pathology, testing companies, and treating physicians to facilitate NGS testing and reporting; (4) referring patients with NGS testing to the Molecular Tumor Board (MTB), which evaluates NGS reports for all patients at the Markey Cancer Center [15,16,17]; and (5) facilitating communication between the MTB and treating physicians.

The primary outcome of this study was to compare the rates of guideline-concordant care NGS testing pre- and post-nurse navigator implementation; therefore, patients were not randomized and reviewers were not blinded. All consecutive patients discussed at the weekly gynecological oncology tumor board were considered for inclusion and were identified using the tumor board database (OncoLens). Each patient’s clinical and demographic data are compiled in the database. 

The study population included female patients aged 18 years old and older with ovarian cancer who received treatment at the Markey Cancer Center between 16 February 2021 to 31 August 2022. All patients must have had a new ovarian cancer tissue diagnosis during the study time period or a change in their cancer status that required previous tissue specimen review (i.e., recurrence). Patients were excluded if they did not have any data in the electronic medical record (*n* = 1), were reviewed at tumor board as a second opinion (*n* = 1), if their primary cancer was not ovarian (*n* = 1), or they had a tumor that was not recommended to have NGS by guidelines (*n* = 18). Histology classification was determined by pathology report. Excluded pathologies were borderline tumors (*n* = 8), granulosa cell tumors (*n* = 4), sex cord-stromal tumors (*n* = 2), liposarcoma (*n* = 1), and benign (*n* = 1). Patients were also excluded if their tissue specimen was reviewed at tumor board between 1 September 2021 and 31 December 2021, as the newly hired nurse navigator was being onboarded. 

After IRB approval, the tumor board database (OncoLens) was used to identify all patients with ovarian cancer reviewed between 16 February 2021 and 31 August 2021 which were defined as prior to the implementation of nurse navigation (pre-NN). A retrospective review was performed to collect the demographic, clinicopathologic, and genetic data from the electronic health record and OncoLens. Patients with ovarian cancer reviewed between 1 January 2022 and 31 August 2022 were defined as post-nurse navigator (post-NN). Most of the post-NN data were collected prospectively during active management of the patients although the electronic health record and OncoLens were also used. 

Patients were either categorized as having no tumor testing, targeted panel tumor testing, or next generation sequencing. The targeted screening panel used at Markey included testing for *ATM*, *BARD1*, *BRCA1*, *BRCA2*, *BRIP1*, *CHEK2*, *MRE11A*, *NBN*, *PALB2*, *RAD51C*, and *RAD51D*, which missed several HRD genes as well as the NCCN recommended LOH, TMB, and evaluation of MMR [5]. MSI status was sent as an add-on test if the genetic counselor ordered testing; however, this was not done for every patient. The rationale for not ordering NGS was also recorded and categorized as appropriately or inappropriately not ordered. Appropriately not ordered rationales were as follows: (1) not enough tissue; (2) the community provider was the primary treating physician; and (3) a patient with two malignancies had NGS on their more advanced, non-ovarian primary after multidisciplinary discussion. Inappropriate rationales were as follows: (1) cancer characteristics (e.g., stage, grade, or type of tumor); (2) unknown; (3) targeted panel testing; and (4) waiting for recurrence.

The mutations identified using targeted panel testing and NGS were summarized and noted to be therapeutically actionable or not actionable mutations in either the frontline or recurrent setting. Frontline actionable mutations were defined as high LOH (≥16), *BRCA1/2*, or HRD genes (*ATM*, *BARD1*, *BRIP1*, *CHEK1*, *CHEK2*, *FANCA*, *FANCC*, *MRE11A*, *NBS1*, *PALB2*, *RAD50*, *RAD51*, *RAD51B*, *RAD51C*, *RAD52*, *RAD54L*) [5,18,19,20,21,22,23,24]. Recurrent actionable mutations were defined as PDL1 ≥1%, *NTRK* gene fusion, *BRAF* mutation, *RET* mutation, MMR deficiency, MSI-high, and TMB-high tumors (>10) [5,25,26,27,28,29]. Finally, patients were also considered to have a recurrent actionable mutation if the molecular tumor board recommended a mutation-specific treatment including recommendations targeting *ERBB2*, *FGFR2-ATE1*, and *AKT2* [15,16,17]. FRα testing was not available at the time of this study. 

Descriptive statistics were used to summarize demographic, clinicopathologic, molecular, and germline testing data, and comparisons before and after nurse navigation implementation were made using the student *t*-test, Chi-square test, and Fisher’s exact test if there were few observations for individual cells (sample size was less than 5). Student *t*-test, Chi-square test, and Fisher’s exact test were also conducted to compare the effect before and after nurse navigator implementation on the number of NGS, the reason for no NGS, timeliness for NGS testing, tumor mutations, germline testing results, and timeliness in ovarian cancer patients. Univariate logistical regressions were used to identify factors associated with NGS testing. Haldane–Anscombe correction was used to fix the odds ratio for data cells with 0 observation by adding 0.5 to each cell. No sample size calculation was performed. 

## 3. Results

### 3.1. Study Population

There were 78 patients included in this study: 34 patients before nurse navigation and 44 patients after implementing nurse navigation. The demographic and clinicopathologic data were similar between groups (Table 1). The mean age at diagnosis was 61.7 years (range 23–89), the mean BMI was 30.43 mg/m^2^ (IQR 12.7), and most patients were Non-Hispanic White (*n* = 73, 94%) and had more than one comorbidity. Most women were diagnosed with primary ovarian cancer (*n* = 62, 79.49%), while 16.67% (*n* = 13) had fallopian tube carcinoma and 3.85% (*n* = 3) had primary peritoneal cancer. The most common histology was high-grade serous carcinoma (*n* = 59, 75.64%), followed by endometrioid carcinoma (*n* = 10, 12.82%), low-grade serous carcinoma (*n* = 4, 5.13%), mixed clear cell and endometrioid (*n* = 2, 2.56%). Finally, one patient each had a mucinous, small cell, and dedifferentiated carcinoma (1.28%). The histology distribution was similar between cohorts (*p* = 0.07), although the post-group had more non-high-grade serous histology (*n* = 15 versus *n* = 3). The cohort had a large distribution of pathologic stages, with the most common stage being IIIC. The type of tumor specimen evaluated for testing was not different between groups (*p* = 0.40), and the specimen for most patients was obtained before treatment with chemotherapy (7.89% pretreatment biopsy and 44.74% primary surgery). The patients in the post-cohort were more likely to be receiving their initial therapy, while the pre-NN were more likely to be recurrent, persistent, or have no evidence of disease at the time of data collection (*p* = 0.0002). 

### 3.2. Clinical Impact

Nurse navigation significantly improved guideline-concordant NGS, increasing testing rates from 35.29% (12/34) to 77.27% (34/44) after navigation, *p* = 0.002 (Table 2). Nurse navigation decreased the guideline-discordant targeted panel testing rate from 35.29% (10/34) to 0% (0/44) after navigation. The rates of no tumor testing also decreased after nurse navigation from 29.41% (10/34) to 22.72% (10/44) after navigation. On univariate analysis, no clinical or provider factors associated with receiving NGS in the pre-NN group. In the post-NN group, patients with a higher stage or two or fewer comorbidities were more likely to receive NGS. The ordering physicians trended towards significance. All other demographic and clinicopathologic factors were not associated with a lack of testing, including age, BMI, race, primary cancer site, histology, and current disease status (Table 3). The rate of NGS was not different between ages <50 yo (9/15, 60%) and ≥50 (37/63, 58.73%), regardless of NN pre/post status (*p* = 0.93). 

The rationale for not ordering NGS varied between the cohorts (*p* = 0.0004) (Table 4). In the pre-NN cohort, the most common reason for not ordering NGS testing was that targeted panel testing was ordered (13/22, 59%), while in the post-NN group, the most common rationale was the cancer characteristics (5/10, 50%). Of the seven patients without NGS due to cancer characteristics, five were stage IA endometrioid histology, one was stage IA low-grade serous, and one was stage IIA mixed clear cell and endometrioid. Inappropriately not ordered NGS were higher in the pre-NN cohort than the post-NN cohort (86.36% versus 50%) and trended towards statistical significance (*p* = 0.0722). 

### 3.3. Timeliness of Testing 

The turnaround time for tumor testing results was significantly reduced after NN implementation (Table 5). The overall total time of testing, defined as the date from specimen collection to date of NGS reporting, was reduced from 145.2 days (95% CI 103.5–187.0) to 42.8 days (95% CI 31.6–54.0) in the post-NN cohort (*p* < 0.0001). NN improved interim time intervals, including the date of specimen collection to the date of NGS order, which was decreased from 102.6 days (95% CI 69.3–136.0) to 24.7 days (95% CI 13.4–35.9) with nurse navigation (*p* < 0.0001). The date the sample was received by the testing company to date resulted was decreased from 23.1 days (95% CI 16.0–30.1) to 12.3 days (95% CI 11.3–13.2) after nurse navigation (*p* = 0.0003). The only time interval not significantly improved was the physician order date to date received by the testing company (6.1 days versus 4.7 days, *p* = 0.48). There was a trend toward more timely testing overall for ages <50 yo than ≥50 yo (44.4 days versus 92.6 days) regardless of NN pre/post status; however, it was not statistically significant (*p* = 0.09). 

### 3.4. Mutational Analysis

There were 154 mutations noted in the entire cohort who received NGS or targeted panel testing, with 107 mutations identified in the post-NN group and 47 in the pre-NN group (Figure 1). Nine patients had no mutations identified, all receiving targeted panel testing in the pre-NN group (Table 6). The post-NN group had a significantly higher rate of tumor mutations found, as 100% of patients (*n* = 34) had a mutation noted versus only 62.5% (15/24) in the pre-NN group (*p* = 0.0001). The rate of actionable mutations in the front-line setting was higher in the post-NN group, 41.2% versus 33.3%; however, it was not statistically significant (*p* = 0.41). The rate of actionable mutations in the recurrent setting was significantly higher in the post-NN group, 67.6% versus 20.8% (*p* = 0.0005). The post-NN group also had a higher rate of LOH testing results, 37.5% versus 97.1% (*p* < 0.0001); while the rate of high LOH (score ≥ 16) in the patients with LOH testing results was not different (44.4% versus 36.4%, *p* = 0.71). Among patients with an actionable mutation who underwent NGS, 88.2% (30/34) would not have been identified by TPT alone. Of those, 40% (12/30) were mutations that would have been missed in the front-line setting, and all are attributed to high LOH scores.

### 3.5. Germline Testing 

There was no difference in physician recommendations for germline testing between the pre-NN cohort, 91.2% (31/34) and the post-NN cohort, 93.2% (41/44); however, the rate of completed germline testing decreased after nurse navigation from 79.4% (27/34) to 50.0% (22/44) (*p* = 0.008) (Table 7). In the entire cohort, 29 patients who did not complete germline testing. Of the seven patients in the pre-NN cohort who did not undergo germline testing, three patients were not recommended for genetic counseling (GC), two declined referral for GC, one did not come to their GC appointment, and one declined testing after their GC appointment. Of the twenty-two patients in the post-NN cohort who did not undergo germline testing, three patients were not recommended for GC, eight had missing GC orders despite physician recommendations, one did not come to their GC appointment, eight never scheduled GC appointments, and two had GC with no completion of testing. Among patients completing genetic testing, the time interval from the physician recommending genetic counseling and germline testing results was significantly decreased in the post-NN group, from 135.4 days (95% CI 88.4–182.5) to 69.1 days (95% CI 34.8–103.5), *p* = 0.03. There was no difference in positive germline findings between the groups (14.8% versus 25%, *p* = 0.49, Table 7). The rate of BRCA mutation or HRD was not different between groups (14.8% versus 16.7%, *p* = 1.00) (Table 7).

## 4. Discussion

Nurse navigation significantly improved guideline-concordant somatic tumor testing rates for patients with ovarian cancer from approximately 35% to 75%. This was primarily accomplished by a physician education session and consensus building, which resulted in a switch from targeted panel testing (which only included *ATM*, *BARD1*, *BRCA1*, *BRCA2*, *BRIP1*, *CHEK2*, *MRE11A*, *NBN*, *PALB2*, *RAD51C*, *RAD51D*, and intermittent MSI test add on) to NGS testing, which expanded the panel of genes tested, and the introduction of a nurse navigator that facilitated NGS testing and communication. The post-NN group had a higher rate of tumor mutations identified and a higher rate of actionable tumor mutations, with 73.5% of patients in the post-NN group harboring an actionable mutation. When evaluating targetable mutations in the front-line setting, more patients had targetable mutations in the post-NN group. However, this did not reach statistical significance, most likely due to the small sample size and the fact that some HRD-related genes were tested in the pre-NN group using a targeted panel. The missed mutations in patients with targeted panel testing could impact their ability to receive maintenance therapy and candidacy for targeted therapies in the recurrence setting, with the potential to impact clinical outcomes.

The reason for not ordering NGS was more likely to be appropriate with nurse navigation, which eliminated several inappropriate reasons including using a targeted panel test, waiting for recurrence, and unknown. Five patients did not receive guideline-concordant testing after nurse navigation due to cancer characteristics related to endometrioid histology and/or low stage. Four total patients in the entire cohort did not have enough tissue for NGS; these likely represented patients with a pretreatment biopsy or a recurrence biopsy, as it is unlikely to not have enough pathologic tissue after cytoreductive ovarian surgery. A previous study suggested that race impacted the rate of molecular testing in ovarian cancer; however, this was not noted in our study [8], most likely due to the small sample size and a primarily Caucasian population. On univariate analysis, individuals with a higher stage or fewer comorbidities were more likely to receive NGS testing. While not reaching statistical significance, test ordering varied by physician, which, taken together, likely suggests variability among physicians of the perceived value of NGS testing in patients with less advanced disease or with more comorbidities.

NGS testing timeliness was improved by over 100 days after nurse navigation, with the time to order the test after specimen collection being 75 days shorter because of nurse navigator coordination. The time from ordering the test to the specimen receipt by the testing company was unchanged, as it was related to the pathology workflow of our facility and not nurse navigation. However, it was reported for completeness. Finally, the timeliness of the testing company reporting results after receiving the specimen was decreased by ten days. This interval is likely inherent to the testing company but could have been improved by nurse navigation and the resolution of specimen testing or reporting issues. There were five different testing companies used in the pre-NN cohort. In contrast, only two were used in the post-NN cohort which may explain the wider confidence intervals in the pre-NN cohort, as each testing company may have a variable turnaround time after the specimen is received.

Despite no difference in physician recommendation for germline genetic testing, the rate of genetic testing completion was lower after nurse navigation, declining from 79.4% to 50%. The temporal design of this study likely impacted the post-NN patients’ ability to have enough time to schedule appointments with genetic counseling (*n* = 8) and complete genetic testing (*n* = 2). However, the pre-NN cohort had a higher rate of targeted panel testing, which was often ordered by the genetic counselor at the same time as germline testing, which may contribute to the higher number of missed genetic counseling orders in the post-NN group (*n* = 8). Despite decreased rates of germline testing in the post-NN cohort, more actionable mutations were found in the post-NN cohort, which is the most important factor for treatment decisions.

There are several strengths to our study, namely the non-biased sampling due to the database use and the prospective data collection in our post-NN group, which minimized confounders affecting our rates of NGS testing. Our study was, however, limited by the temporal design of the study, with the pre-nurse navigation cohort having more time to complete both germline and somatic testing. Despite no difference in the type of specimen used for testing (pre-treatment, post-treatment, or recurrence), the post-nurse navigation group was more likely to still be on their initial therapy, which impacted our ability to report the rate of targeted therapies received and the survival outcomes. However, it is well documented in the literature that targeted therapies for actionable mutations are associated with an improved progression-free survival and response rate compared to conventional chemotherapy [16,30,31]. Furthermore, we defined front-line actionable mutations by combining established NCCN guidelines with the genes associated with HRD, as HRD genes are thought to confer responses to PARP inhibitors [5,18,19,20,21,22,23,24]. Mutation in non-*BRCA* HRD genes is rare (ranging from 0.2–2%); therefore, each gene is difficult to study individually; however, most are also associated with high LOH [32]. Only five patients had a mutation in a non-*BRCA* HRD gene with low or unknown LOH (*PALB2*-LOH 9%, *PALB2*-LOH 3%, *CHEK2*-LOH 12%, and two *ATM* with unknown LOH). Our definition of a recurrent actionable mutation was a combination of NCCN guidelines, PDL1 > 1%, and Molecular Tumor Board recommendations [5,15,16,17,25,26,27,28,29]. Despite our small sample size, the odds ratio for receiving NGS testing is 6.23 times higher for patients after implementation; with a sample size of 78 and a 0.05 significance level, our primary objective has 0.95 power; however, the additional exploratory analysis of our cohorts was underpowered. Furthermore, the small sample size limits our ability to demonstrate minor differences and may account for the low numbers of clear cell and endometrioid cancer. A single institution design limits the generalizability of our outcomes; however, we anticipate that education, consensus building, and nurse navigation could improve NGS testing in many institutions.

This project identified five patients who did not receive NGS after nurse navigation due to their perceived cancer characteristics. This discovery provided an opportunity to re-educate our providers and nurse navigator. Furthermore, we are considering implementing liquid biopsies into our practice for patients with insufficient tissue for NGS. Although liquid biopsies are not the standard of care for ovarian cancer, there is promise for using them to track disease and mutational status after recurrence, which likely describes the patient population who did not have enough tissue for NGS. A quality improvement project is underway to improve our germline testing completion rate. Although a germline testing completion rate of 50% is higher than the national average of 30%, it is still not concordant with guidelines and could impact patients’ treatment decisions and family testing [9].

We believe the rate and timeliness of testing were directly impacted by nurse navigation. On discussion with the nurse navigator, they think their nursing background helped them understand who was a candidate for testing, troubleshoot, and feel confident in recommending testing to the physician if the physician did not recommend testing when indicated. Our nurse navigator has several other responsibilities within our cancer center; however, if this is the only responsibility, a non-medically trained clinic staff member could potentially obtain similar results if education is provided.

## 5. Conclusions

Many studies have shown nurse navigation improves outcomes and timeliness of cancer care [11,12,13,14]. However, our study is the first to show the benefit of nurse navigation of somatic testing in gynecological clinics. Somatic tumor testing is paramount when deciding candidacy for maintenance therapy and recurrent therapy. Our study reveals that tasking a nurse navigator to attend tumor boards and coordinate test ordering and results can improve testing rates and timeliness. Furthermore, we showed that discontinuing targeted panel testing in favor of NGS can reveal actionable mutations that would have been missed with targeted panel testing alone.

## Figures and Tables

**Figure 1 cancers-15-03192-f001:**
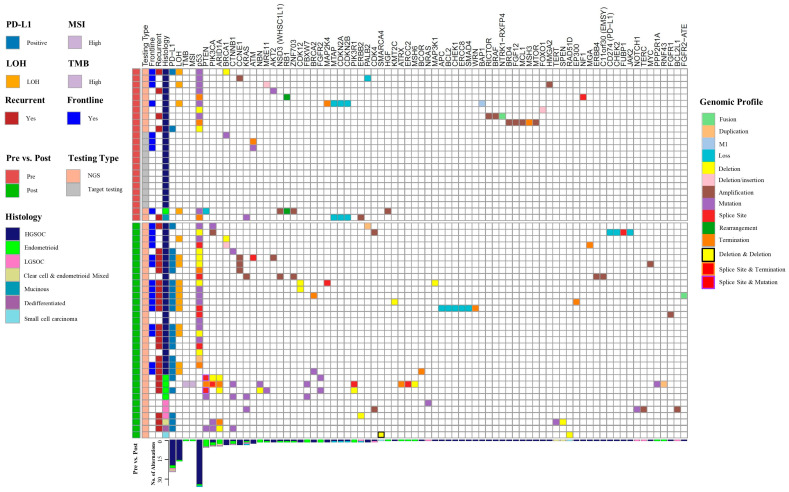
Genomic profiles and histology of patients included in the study. Columns demonstrate Pre- vs. Post-NN group, the testing type used, the actionable mutation in frontline versus recurrent, histology, genes, and biomarkers. Rows demonstrate individual tumor profiles. LOH, loss of heterozygosity; MSI, microsatellite instability; PD-L1: programmed death-ligand 1; TMB, tumor mutational burden. The genomic profile describes the type of mutation in each gene; deletion and deletion denotes two deletion mutations within one gene; splice site and termination denotes a splice site and termination mutation within one gene; splice site and termination denotes a splice site and termination mutation within one gene.

**Table 1 cancers-15-03192-t001:** Baseline Demographic and Disease Characteristics in Ovarian Cancer Patients Pre and Post Implementation of a Nurse Navigator.

	All	Pre NN	Post NN	*p*-Value *
	*n* = 78	*n* = 34	*n* = 44	
Age at diagnosis, year, mean (range)	61.67 (23–89)	63.76 (32–89)	60.05 (23–82)	0.22
Age at diagnosis, *n* (%)				0.14
<50 years old	15 (19.23)	4 (11.76)	11 (25)	
≥50 years old	63 (80.77)	30 (88.24)	33 (75)	
BMI, mean (IQR)	30.43 (12.7)	29.78 (11.6)	30.93 (14.8)	0.57
Race/Ethnicity, *n* (%)				0.43
Non-Hispanic White	73 (93.59)	31 (91.81)	42 (95.45)	
Black	2 (2.56)	1 (2.94)	1 (2.27)	
Hispanic	2 (2.56)	2 (5.88)	0	
Unknown	1 (1.28)	0	1 (2.27)	
Primary, *n* (%)				0.81
Ovary	62 (79.49)	26 (76.47)	36 (81.82)	
Fallopian	13 (16.67)	7 (20.59)	6 (13.64)	
Peritoneal	3 (3.85)	1 (2.94)	2 (4.55)	
Histology, *n* (%)				0.07
HGSOC	59 (75.64)	30 (88.24)	29 (65.91)	
LGSOC	4 (5.13)	0	4 (9.09)	
Endometrioid	10 (12.82)	2 (5.88)	8 (18.18)	
Mucinous	1 (1.28)	1 (2.94)	0	
Clear cell and endometrioid mixed	2 (2.56)	1 (2.94)	1 (2.27)	
Dedifferentiated	1 (1.28)	0	1 (2.27)	
Small cell carcinoma	1 (1.28)	0	1 (2.27)	
Initial Stage, *n* (%)				0.40
IA	14 (17.95)	4 (11.76)	10 (22.73)	
IB	1 (1.28)	0	1 (2.27)	
IC1	3 (3.85)	2 (5.88)	1 (2.27)	
IC2	2 (2.56)	1 (2.94)	1 (2.27)	
IC3	2 (2.56)	2 (5.88)	0	
IIA	2 (2.56)	2 (5.88)	0	
IIB	6 (7.69)	1 (2.94)	5 (11.36)	
IIIA1i	1 (1.28)	1 (2.94)	0	
IIIA1ii	1 (1.28)	0	1 (2.27)	
IIIA2	1 (1.28)	0	1 (2.27)	
IIIB	1 (1.28)	0	1 (2.27)	
IIIC	28 (35.90)	13 (38.24)	15 (34.09)	
IVA	8 (10.26)	4 (11.76)	4 (9.09)	
IVB	7 (8.97)	4 (11.76)	3 (6.82)	
Specimen, *n* (%)				0.40
Pretreatment biopsy	6 (7.89)	2 (5.88)	4 (9.52)	
Primary surgery	34 (44.74)	17 (50)	17 (40.48)	
Interval debulking	13 (17.11)	3 (8.82)	10 (23.81)	
Recurrence	5 (6.58)	2 (5.88)	3 (7.14)	
Unknown	18 (23.68)	10 (29.41)	8 (19.05)	
Current Disease Status, *n* (%)				0.0002
Initial therapy	34 (43.59)	6 (17.65)	28 (63.64)	
Recurrent	11 (14.10)	7 (20.59)	4 (9.09)	
Persistent	7 (8.97)	6 (17.65)	1 (2.27)	
No evidence of disease	25 (32.05)	14 (41.18)	11 (25)	
MD, *n* (%)				0.77
1	17 (21.79)	8 (23.53)	9 (20.45)	
2	8 (10.26)	3 (8.82)	5 (11.36)	
3	8 (10.26)	4 (11.76)	4 (9.09)	
4	22 (28.21)	9 (26.47)	13 (29.55)	
5	11 (14.10)	3 (8.82)	8 (18.18)	
6	12 (15.38)	7 (20.59)	5 (11.36)	
Hypertension, *n* (%)				0.24
No	31 (39.74)	11 (32.35)	20 (45.45)	
Yes	47 (60.26)	23 (67.65)	24 (54.55)	
Diabetes, *n* (%)				0.50
No	59 (75.64)	27 (79.41)	32 (72.73)	
Yes	19 (24.36)	7 (20.59)	12 (27.27)	
Cardiac, *n* (%)				0.51
No	64 (82.05)	29 (85.29)	35 (79.55)	
Yes	14 (17.95)	5 (14.71)	9 (20.45)	
Respiratory, *n* (%)				0.56
No	62 (79.49)	26 (76.47)	36 (81.82)	
Yes	16 (20.51)	8 (23.53)	8 (18.18)	
Neuro, *n* (%)				1.00
No	74 (94.87)	32 (94.12)	42 (95.45)	
Yes	4 (5.13)	2 (5.88)	2 (4.55)	
Renal, *n* (%)				0.65
No	73 (93.59)	31 (91.18)	42 (95.45)	
Yes	5 (6.41)	3 (8.82)	2 (4.55)	
Total Comorbidities, *n* (%)				0.88
0	20 (25.64)	7 (20.59)	13 (29.55)	
1	26 (33.33)	13 (38.24)	13 (29.55)	
2	19 (24.36)	8 (23.53)	11 (25)	
3	11 (14.10)	5 (14.71)	6 (13.64)	
4	2 (2.56)	1 (2.94)	1 (2.27)	

* *p*-values were calculated with *t*-test for numerical variable, Ch-square test for categorical variable, and Fisher’s exact test if *n* < 5, significance level: 0.05.

**Table 2 cancers-15-03192-t002:** Next Generation Sequencing in Ovarian Cancer Patients Before and After Nurse Navigator Implementation.

	Pre-NN (*n* = 34)	Post-NN (*n* = 44)	*p*-Value *
NGS, *n* (%)	12 (35.29)	34 (77.27)	0.0002
No NGS, *n* (%)	22 (64.71)	10 (22.72)	
No testing	10 (29.41)	10 (22.72)	
Targeted panel testing	12 (35.29)	0	

* *p*-values were calculated with Chi-square test.

**Table 3 cancers-15-03192-t003:** Logistic Regression-Estimated Odds Ratios (OR) and 95% Confidence Intervals (CI) Associated with Receiving NGS Testing.

Demographic	Pre NN N = 34		Odds Ratio *^#^*	95% CI	*p*-Value	Post NN N = 44		Odds Ratio *^#^*	95% CI	*p*-Value
	NGS-No *n* = 22	NGS-Yes *n* = 12				NGS-No *n* = 10	NGS-Yes *n* = 34			
Age, mean (SD)	62.27 (12.94)	66.5 (9.46)	1.03	0.97–1.10	0.32	61.4 (11.12)	59.65 (15.15)	0.99	0.94–1.04	0.73
Age, *n*			6.08	0.30–123.2	0.24			0.70	0.12–3.90	0.68
<50 yo	4	0 *				2	9			
≥50 yo	18	12				8	25			
BMI, mean (SD)	29.49 (7.93)	30.32 (8.97)	1.01	0.93–1.10	0.78	31.75 (9.49)	30.69 (9.33)	0.99	0.92–1.07	0.75
MD					0.71					0.09
1 (reference)	5	3				2	7			
2	2	1	0.83	0.05–13.63		3	2	0.24 *	0.03–2.07	
3	3	1	0.56	0.04–8.09		2	2	0.33 *	0.04–3.21	
4	4	5	2.08	0.30–14.55		0	13	9.00 *	0.38–213.17	
5	2	1	0.83	0.05–13.63		0	8	5.67 *	0.23–137.80	
6	6	1	0.28	0.02–3.58		3	2	0.24 *	0.03–2.07	
Total Comorbidities					0.41					0.03
0–2 (reference)	19	9				6	31			
>2	3	3	2.11	0.35–12.59		4	3	0.16	0.03–0.90	
Primary					0.88					0.46
Ovary (reference)	17	9				9	27			
Fallopian/ Peritoneal	5	3	1.13	0.22–5.86		1	7	2.33	0.25–21.63	
Histology					0.52					0.23
HGSOC (reference)	20	10				5	24			
Other	2	2	2.00	0.24–16.36		5	10	0.42	0.10–1.76	
Initial Stage					0.11					0.03
Early (reference)	10	2				7	11			
Late	12	10	4.18	0.74–23.61		2	23	6.13	1.20–31.33	
Specimen					0.30					0.95
Pre-treatment biopsy (reference)	0	2				0	4			
Primary surgery	11	6	0.11 *	0.01–2.73		0	17	3.89 *	0.07–224.22	
Interval debulking	1	2	0.33 *	0.01–12.82		0	10	2.33 *	0.04–136.98	
Recurrence	0	2	1.00 *	NA		0	3	0.78 *	0.01–49.9	
Unknown	10	0	NA	NA		8	0	NA	NA	
Current Disease Status					0.32					0.64
Initial therapy (reference)	3	3				7	21			
NED	11	3	0.27	0.04–2.11		3	8	0.85 *	0.19–3.79	
Recurrent/Persistent	7	6	0.86	0.12–5.94		0	5	3.84 *	0.19–78.00	

^#^ Odds ratios were calculated with logistical regression; * Haldane–Anscombe correction was used to fix the odds ratio for data cells with 0 observation by adding 0.5 to each cell.

**Table 4 cancers-15-03192-t004:** Rationale for No NGS.

Reason	Pre-NN	Post-NN	*p*-Value *
	*n* = 22	*n* = 10	
Reason for no NGS, *n* (%)			0.0004
Targeted screening	13 (59.08)	0	
Unknown	3 (13.64)	0	
Cancer characteristics (e.g., stage/grade/type)	2 (9.09)	5 (50)	
Community provider primary	2 (9.09)	1 (10)	
Not enough tissue	1 (4.55)	3 (30)	
Wait for recurrence	1 (4.55)	0	
Sent on second primary malignancy	0	1 (10)	
Not ordered (appropriate reason) **, *n* (%)	3 (13.64)	5 (50)	0.0722
Not ordered (inappropriate reason) ***, *n* (%)	19 (86.36)	5 (50)	

* *p*-values were calculated with Chi-square or Fisher’s exact tests; ** Appropriate reasons: not enough tissue, community provider primary, and NGS sent on more advanced second primary; *** Inappropriate reasons: stage, grade or type of tumor, unknown, targeted screening ordered, and wait for recurrence.

**Table 5 cancers-15-03192-t005:** Timeliness of NGS Testing Pre and Post Nurse Navigation.

Time Interval	Pre NN (*n* = 24)	Post NN (*n* = 34)	*p*-Value *
	*n* = 23	*n* = 34	<0.0001
Specimen collection date to date resulted ***, mean number days (95% CI)	145.2 (103.5–187.0)	42.8 (31.6–54.0)	
	*n* = 22	*n* = 33	<0.0001
Specimen collected date to physician order date, mean number days (95% CI)	102.6 (69.3–136.0)	24.7 (13.4–35.9)	
	*n* = 21	*n* = 33	0.48
Physician order date to date received by testing company **, mean number days (95% CI)	6.1 (2.3–10.0)	4.7 (4.7–6.1)	
	*n* = 22	*n* = 34	0.0003
Date received by testing company ** to date resulted ***, mean number days (95% CI)	23.1 (16.0–30.1)	12.3 (11.3–13.2)	

* *p*-values were calculated with *t*-test; ** Date received by the testing company is defined as the day the testing company received the specimen; this date is located on the test report or on the company’s patient portal; *** Date resulted is defined as the day the testing company supplied the results, this date is located on the test report.

**Table 6 cancers-15-03192-t006:** Summary of Tumor Mutations.

	Pre NN *n* = 24	Post NN *n* = 34	*p*-Value *
Tumor mutation, *n* (%)			0.0001
Yes	15 (62.5)	34 (100)	
No	9 (37.5)	0	
Frontline actionable mutations **, *n* (%)			0.41
Yes	8 (33.3)	15 (44.1)	
No	16 (66.6)	19 (55.8)	
Recurrence actionable mutations ***, *n* (%)			0.0005
Yes	5 (20.8)	23 (67.6)	
No	19 (79.2)	11 (32.4)	
LOH results, *n* (%)			<0.0001
Yes	9 (37.5)	33 (97.1)	
No	15 (62.5)	1 (2.9)	
LOH High (score ≥ 16), *n* (%)	*n* = 9	*n* = 33	0.71
Yes	4 (44.4)	12 (36.4)	
No	5 (55.6)	21 (63.6)	

* *p*-values were calculated with Chi-square or Fisher exact tests; ** Frontline actionable mutations: high LOH, *BRCA1/2*, or HRD (*ATM*, *BARD1*, *BRIP1*, *CHEK1*, *CHEK2*, *FANCA*, *FANCC*, *MRE11A*, *NBS1*, *PALB2*, *RAD50*, *RAD51*, *RAD51B*, *RAD51C*, *RAD52*, *RAD54L*); *** Recurrent actionable mutations: PDL1 positive, *NTRK* gene fusions, *BRAF* mutation, dMMR, *RET*, MSI-high, and TMB-high tumors (>10), and molecular tumor board-recommended actionable *ERBB2*, *FGFR2-ATE1*, and *AKT2* mutations.

**Table 7 cancers-15-03192-t007:** Germline Testing Results and Timeliness Pre- and Post-Nurse Navigation.

	**Pre NN**	**Post NN**	***p*-Value ***
	*n* = 34	*n* = 44	
Physician recommended germline testing, *n* (%)			1.00
Yes	31 (91.2)	41 (93.2)	
No	3 (8.8)	3 (6.8)	
Germline testing completed, *n* (%)			0.008
Yes	27 (79.4)	22 (50)	
No	7 (20.6)	22 (50)	
	*n* = 27	*n* = 22	
Germline testing results, *n* (%)			0.31
Negative/VUS	23 (85.2)	16 (72.7)	
Positive	4 (14.8)	6 (27.3)	
Germline *BRCA* or HRD mutated **			1.00
Yes	4 (14.8)	4 (18.2)	
No	23 (85.2)	18 (81.8)	
	*n* = 4	*n* = 6	
Positive germline mutation results, *n* (%)			0.30
*ATM*	1 (25)	0	
*BRCA1*	3 (75)	1 (16.7)	
*BRCA2*	0	2 (33.3)	
*NF1*	0	1 (16.7)	
*PALB2*	0	1 (16.7)	
*WRN*	0	1 (16.7)	
	*n* = 6	*n* = 4	
Germline VUS, *n* (%)			0.22
*ATM* VUS	0	2 (50)	
*CDKN1B* VUS	1 (16.7)	0	
*MSH6* VUS	1 (16.7)	0	
*PMS2* VUS	0	1 (25)	
*DICER1* VUS	2 (33.3)	0	
*PALB2* VUS	1 (16.7)	0	
*POLD1* VUS	1 (16.7)	0	
*SMARCA4* VUS	0	1 (25)	
Time Interval Date physician recommended genetics to germline test results, mean number days (95% CI)	*n* = 19 135.4 (88.4–182.5)	*n* = 16 69.1 (34.8–103.5)	0.03
Date physician recommended genetics to genetic counseling, mean number days (95% CI)	*n* = 19 82.5 (43.4–121.7)	*n* = 18 48.4 (17.0–79.8)	0.16
Date of genetic counseling to germline test results, mean number days (95% CI)	*n* = 19 52.9 (12.3–93.5)	*n* = 15 24.2 (11.9–36.5)	0.21

* *p*-values were calculated using Chi-square or Fisher exact test; ** HRD genes: *ATM*, *BARD1*, *BRIP1*, *CHEK1*, *CHEK2*, *FANCA*, *FANCC*, *MRE11A*, *NBS1*, *PALB2*, *RAD50*, *RAD51*, *RAD51B*, *RAD51C*, *RAD52*, *RAD54L*.

## Data Availability

Anonymized data sets may be available upon request. Requests for access to data may be obtained by contacting the corresponding author.

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
