# Peer review of "Implementation of Nurse Navigation Improves Rate of Molecular Tumor Testing for Ovarian Cancer in a Gynecologic Oncology Practice"

_cancers, 2023, doi:10.3390/cancers15123192_

Round 1

Reviewer 1 Report

Rives at el. investigated the compliance with recommended genetic testing in ovarian cancer patients in 7-8months time frame before and after implementing a nurse navigator. Overall, they could show that the education of clinical and nursing staff results in higher compliance and higher rate of genetic testing down in OvCa patients.

The manuscript lacks the following information or must address:
- clear explanation/definition of OvCa histology stratification (the numbers seem not very representative as very low number of endometrioid and no clear cell carcinoma cases were reported)

- analysis specifically comparing the mutation covered by the target panel vs NGS and consider re-testing those pre NN cases with NGS

- definition what is considered an actionable mutation - no citation to any other publication was made to identify those

- discuss the bias based on the amount of recurrent cases in pre NN cohort

- evaluation whether the analyses done are appropriately powered with those limited sample number, amongst others when considering comorbidities/stage in the non-NGS group

- discuss why you see different amount of actionable mutation in the pre and post NN when NGS was performed. One would assume that mutation frequency would be similar in both cohort, otherwise this is a sign that you are dealing with inherent different cohort populations with potential different founder artifacts. The pointed out difference in recurrence already bares a co-founder effect.

- table 5 please define what is meant with date received and date resulted. Furthermore, please add n for those analysis. What is the reasoning behind evaluating factors that are clearly not at all influenced by the implimentation of NN?

- What is the added information/benefit when combining frontline with recurrent mutation status (table 6) - you are washing down any potential seen effects based in the recurrent actionable mutation found.

- figure 1 what is the difference between "deletion" - "deletion & deletion" - "deletion/Insertion"; "splice site" - "splice site &termination/mutation". Please define.

-discuss positive germline mutation found in the pre vs post cohort - frequency differences

-what are the reason for lower germline testing outside of time, different amount of referals, etc?

- is it even cost effective to perform NGS if target panel analysis would provide the same information as NGS? Not all so called actionable mutation found have any real-life consequences in regard to therapeutic options and treatment outcomes. Are there even any clinical implications?

- There is no mentioning of the pre-treatment status of the tissue evaluated. Multiple studies have shown that the mutational background of a tumor substantially changes with neo-adjuvant therapy and those sample that have undergone neoadj. therapy should be analyzed seperately when considering (actionable) pathogenic mutation -  the same is true for recurrent tumors. The molecular setup of a tumor from relapse in a fast amount of cases is different from the initial primary tumor.

n/a

Reviewer 2 Report

The manuscript “Implementation of nurse navigation improves rate of molecular tumor testing for ovarian cancer in a gynecologic oncology practice” by Rives et al. assesses and presents the impact of implementing a nurse navigator to improve the rates and timeliness of molecular tumor testing. They included 78 patients with ovarian carcinoma and they demonstrated the rates of NGS testing, tumor testing, and germline testing. The number of cases that were analyzed is small. Overall, this manuscript yields sufficient novel information or insight into the role of nurse navigation in clinical practice and especially in the rates of molecular tumor testing for ovarian cancer. However, this paper could be accepted after minor revisions.

Minor points:

Introduction:

Line 76: add the references (a) doi: 10.1002/cncr.28024 and (b) doi: 10.1200/OP.20.00899.

Results:

Line 169: add the range of age at cancer diagnosis because you mentioned only the mean age

How many patients were diagnosed with ovarian cancer under 50 years old?

Line 188: clinical impact – did you notice if the age of the cancer diagnosis of a patient has an impact on the rates and timeliness of testing. For example, is there a difference if you have two groups of patients (a) <50 years old and (b) >50 years?

Conclusions:

Line 362: add the references (a) doi: 10.1002/cncr.28024 and (b) doi: 10.1200/OP.20.00899.

Reviewer 3 Report

Thanks for interesting paper, going generalised increase in testing uptake with  the uses of nurse navigators vs physicians. Were there any educational or auditing initiatives done prior to institution of the navigator? i.e. is the nurse truly necessary?
